# Immune Efficacy of the EV71 Vaccine in Fujian Province, China: A Real-World Analysis of HFMD

**DOI:** 10.3390/vaccines11050944

**Published:** 2023-05-04

**Authors:** Junrong Li, Fangqin Xie, Guangcan Lin, Dongjuan Zhang

**Affiliations:** Fujian Provincial Center for Disease Control and Prevention, Fuzhou 350012, China; junrong_li@126.com (J.L.); okxfq@163.com (F.X.); 264970248@163.com (G.L.)

**Keywords:** hand, foot and mouth disease, EV71 vaccine immunization, epidemiological characteristics, changing trend

## Abstract

EV71 vaccine immunization mainly protects the human population against severe and fatal HFMD and has a positive effect on reducing the overall incidence rates of HFMD and of hospitalized cases. In the analysis of data collected over 4 years, we compared HFMD’s incidence rate, severity, and etiological changes in a target population before and after vaccine intervention. The incidence rate of HFMD decreased from 39.02‰ in 2014 to 11.02‰ in 2021, with a decrease rate of 71.7%, and the decrease was statistically significant (*p* < 0.001). The number of hospitalized cases decreased by 68.88%, the number of severe cases dropped by 95.60% and the number of deaths dropped to 0. The proportion of cases caused by the EV71 virus in different populations decreased significantly after the intervention, specifically, by 68.41% among individuals 0–4 years of age, by 74.32% among kindergarten children, by 86.07% in severe cases and by 100% with respect to the number of deaths.

## 1. Introduction

Hand, foot and mouth disease (HFMD) is an acute infectious disease caused by a variety of enteroviruses. Severe cases of HFMD are mostly caused by enterovirus type 71 (EV71) infection, which often occurs in children under 5 years of age for whom it is dangerous, causing high mortality. Human enterovirus rarely causes gastrointestinal symptoms, such as gastroenteritis, but does cause diseases with a variety of clinical symptoms. Children infected with human enterovirus are susceptible to hand, foot and mouth disease. There is a certain probability of aseptic meningitis, meningoencephalitis, acute respiratory diseases, acute myocarditis and acute delayed paralysis. In addition, a small number of children may suffer from complications such as myocarditis, pulmonary edema and aseptic meningoencephalitis. If the disease develops rapidly, as in severe cases, HFMD may lead to death [1]. It has become the main cause of public health emergencies due to acute infectious diseases in various regions of China since 2008 [2]. Therefore, in 2008, HFMD was listed in category C of infectious diseases on the list of reportable diseases in China [3,4,5]. In 2010, the EV71 vaccine was developed [6], and it was put on the market in 2016. In 2017, Fujian Province included the EV71 vaccine in non-EPI vaccine management, and severe cases of HFMD were effectively controlled after this initiative [7]. This study intended to analyze the changes in the epidemiological characteristics of HFMD after inoculation with the EV71 vaccine in Fujian Province and to show evidence of the EV71 vaccine’s effect after the initial inoculation by analyzing the changes in the pathogenesis and etiology of HFMD in order to provide a reference for vaccination decision making.

## 2. Materials and Methods

### 2.1. Research Objects

(1)Target population of the study. During the monitoring period, from 2014 to 2021, the population of individuals under the age of 15 years (0–14 years old) living in Fujian Province was included as the target population of this study: a total of 23,658,364 people were observed every year. With the introduction of the EV71 vaccine into non-EPI vaccine management in Fujian Province in 2017, there were 2,177,931 doses of EV71 vaccine for children under 5 years of age in Fujian from 2018 to 2021, and the whole-course vaccination rate of 2 doses was 54.82%. A group design was adopted. The target population for the 4 years after the EV71 vaccine intervention (2018–2021) was the intervention group, which comprised 206,536 cases; the target population for the 4 years before the vaccine intervention (2014–2017) was the non-intervention group, with 578,779 cases.(2)Outcome variables. During the monitoring period, we evaluated the number of hospitalized cases, severe cases, and deaths by HFMD, as well as the etiological types and the proportion of EV71 infections among laboratory-confirmed cases in the target population, which comprised individuals who were diagnosed with HFMD by healthcare institutions.

### 2.2. Sources of Disease Surveillance Data

Information on individual cases of HFMD was obtained from the infectious-disease-reporting information management system of the “China Disease Prevention and Control Information System” (an internet direct reporting system), and the information regarding the target population of the study was obtained from the “Basic Information System” of the internet direct reporting system. The network direct reporting system is updated by the doctors of township hospitals, maternal and child healthcare hospitals and hospitals within 24 h after a disease is diagnosed as HFMD, and the network direct reporting and case report card are completed. If there is any modification or deletion, reasons are provided. The reported information must be reviewed at the county, city, and provincial CDC levels. In addition, the disease control and prevention department will check the inpatient records, outpatient cases and case report cards many times for each health center and hospital within one year to determine whether there is a missing or wrong report and issue the corresponding report. Report variables include the name, gender, date of birth, contact telephone number, address of current residence, population classification, case classification, date of onset, diagnosis time, date of death, disease name, disease type before revision, diagnosis time before revision, final review time before revision, doctor filling card, doctor filling card date, reporting unit area code, reporting unit, unit type, report card entry time and record card user, unit to which the card-recording user belongs, county and district audit time, city and prefecture audit time, province and city audit time, audit status, revised report time, revised final audit time, final audit death time, revised user, unit to which the revised user belongs, etc. It can also be seen from the reported variables that the network direct reporting system is relatively complete and can guarantee the authenticity of the data.

### 2.3. Analysis Indicators

#### 2.3.1. Incidence Rate and Severity

(1)Incidence rate. Incidence density (ID) = number of HFMD incidents or deaths during the observation period/the sum of person-years in the observation period × 1000. This reflects the incidence rate of HFMD during the monitoring and observation period.(2)The proportion (%) of hospitalized cases, severe cases and death cases, reflecting the severity of HFMD during the monitoring and observation period.

#### 2.3.2. Evaluation of the Vaccine Immunization Effect

(1)Etiological distribution, including the dominant virus strain (%) and the proportions (%) of hospitalized cases, severe cases and death cases caused by EV71. The efficacy of the vaccine was preliminarily evaluated by the calculating the number of EV71 cases before and after the EV71 vaccine intervention.(2)Protection ratio (PR): $\mathrm{PR}=\frac{I_{0}-I_{e}}{I_{0}}\times 100\%$. I_e_ and I_0_ represent the annual incidence rate of HFMD in the intervention group and the non-intervention group, respectively. Combined with the EV71 pathogen proportion, it reflects the ratio of protection against HFMD or death in the EV71 vaccine immune intervention population relative to the non-immunized population.(3)Efficacy Index (IE $=\frac{I_{0}}{I_{e}}$). The IE indicates the proportion of incidence or the mortality rate in the non-intervention group and the intervention group and indirectly reflects the effect of the EV71 vaccine on the immunized population.

### 2.4. Statistical Inference Methods

The frequency data were inferred by X^2^ or trend X^2^, and *p* < 0.05 was considered statistically significant. A protection ratio (PR) ≥ 50% indicated that the vaccine was effective. The correlation or dependence between two quantitative variables were analyzed [8]. (1) A simple linear regression analysis was used to obtain a straight line passing through the origin of the rectangular coordinate system, and the *t* test or *F* test was used to compare the difference between the slope and 0; if *p* < 0.05, the correlation between the two variables was statistically significant. (2) A Pearson correlation analysis was used to obtain R^2^ values ≥ 0.5, indicating that the correlation between the two quantitative variables was highly linear, and the result was meaningful. The conversion formula for the Box–Cox skewness data used for the incidence [9], (Z $=\;\frac{Y^{\lambda }}{\lambda }$, λ ≠ 0), was gradually adjusted and determined by the Kolmogorov–Smirnov normality test. By comparing the maximum absolute difference between the cumulative distribution function of the sample data and the theoretically assumed cumulative distribution function of normal distribution, we used the K-S normal test to test whether the samples in this study conformed to the normal distribution so as to determine the validity of the sampling information.

### 2.5. Quality Control

Fujian Province has long implemented an infectious disease underreporting and reporting quality inspection system, as well as a population and system code maintenance review meeting system. Every year, the health administration department of Fujian Province organizes an investigation on the underreporting and reporting quality of infectious diseases, proactively reviews the registration forms of all medical institutions’ visits and inpatient cases and, in a timely manner, makes a supplementary report if any cases of underreporting are found. In addition to the duplicate inspection of case records within a natural year in the online direct reporting system, the duplicate inspection and data cleaning of aggregated databases were carried out again according to the “Current address GB”, “Name”, “Gender”, “Date of birth”, four indicators across annual case records to ensure the accuracy of monitoring and observation records.

## 3. Results

### 3.1. Comparison of the Incidence Rate before and after the Vaccine Intervention

During the monitoring period, from 2008 to 2021, the target population (0–14 years old) of the HFMD monitoring project in Fujian Province included 23,658,364 individuals, the incidence rate was 24.46‰, the number of hospitalized cases was 42,654, accounting for 7.37%, the number of severe cases was 0.18%, 23 subjects died and the mortality rate was 0.10/100,000. As reported in Table 1, the EV71 vaccine intervention group (12,165,153 person-years) and the non-intervention group (11,493,211 person-years) showed a statistically significant difference in the number of cases (X^2^ = 58,807.25, *p* < 0.001); there were statistical differences in the three proportions of hospitalized cases, severe cases and death cases between the intervention group and the non-intervention group (X^2^ = 499.66, *p* < 0.001).

The protection ratio (PR) and efficacy index (IE) were used to preliminarily evaluate the population immunity effect of the EV71 vaccine. The statistical analysis showed that the protection ratio of the two groups before and after EV71 vaccine intervention was 47.58%, and the efficacy index was 1.91; the protection ratio against hospitalization was 41.36%, and the efficacy index was 1.71; the protection ratio against severe disease was 94.64%, and the efficacy index was 18.64; the protection ratio against death was 95.71%, and the efficacy index was 23.29. It is suggested that for diseases with multiple pathogens such as HFMD, the goal of vaccine immunization—EV71, in this case—is to protect against severe and fatal cases.

From 2014 to 2021, the number of HFMD cases in individuals aged 0–4 years in Fujian Province was 529,703, accounting for 91.52% of the total cases; the incidence rate was 54.05‰. The number of hospitalized cases was 39,606, accounting for 92.85% of the total number of hospitalizations, while the number of severe cases was 1021, accounting for 96.23% of the total number of severe cases; 22 people died, accounting for 95.65% of the total number of deaths. There was a statistically significant difference in the incidence rate between the EV71 vaccine intervention group and the non-intervention group (X^2^ = 62,243.89, *p* < 0.001); there were significant differences in the three proportions of hospitalized cases, severe cases and death cases between the intervention group and the non-intervention group (X^2^ = 454.79, *p* < 0.001). The statistical analysis showed that the protection ratio with respect to the incidence rates in the two groups aged 0–4 years before and after the EV71 vaccine intervention was 49.58%, and the efficacy index was 1.99. In addition, the protection ratio with respect to the hospitalization indicator was 44.20%, and the efficacy index was 1.79; the protection ratio with respect to the number of severe cases indicated was 94.81%, and the efficacy index was 19.27; the protection ratio with respect to death was 95.57%, and the efficacy index was 22.59, as reported in Table 2.

Compared with the non-intervention group, the HFMD incidence rate in the EV71 vaccine intervention group aged 0–4 years in each district and city decreased significantly, and the difference in the incidence rate between the two groups was statistically significant (X^2^ = 33,934.78, *p* < 0.001); the protection ratio against HFMD fluctuated between 24.77 and 60.43%. Compared with the non-intervention group, the number of severe cases and deaths in the intervention group also decreased significantly. The protection ratio against severe disease and death in each district and city was generally higher than 88.65%, as shown in Table 3.

### 3.2. Changes in Pathogenic Characteristics before and after the Vaccine Intervention

During the surveillance period, from 2014 to 2021, there were 24,779 individuals with laboratory-confirmed cases of HFMD, all of whom were hospitalized. Of these cases, 0–4-year-old children accounted for 92.96%; the ratio of males to females was 1.67:1; children living at home accounted for 81.15%; severe and fatal cases accounted for 3.12%; cases in cities and towns accounted for 61.68%, as shown in Table 4. Compared with the non-intervention group, the proportion of EV71 virus infections in the intervention group decreased by 68.28%, the proportion of CoxA16 infections decreased by 0.86% and the proportion of other enterovirus infections increased by 36.81%. The differences in the proportions of virus types before and after the intervention were statistically significant (X^2^ = 1459.52, *p* < 0.001). In different population groups and layers, the proportions of infections with the EV71 virus decreased significantly before and after the intervention, specifically, by 68.41% in individuals 0–4 years of age, by 74.32% in nursery children, by 86.07% with respect to severe cases and by 100% with respect to deaths. The proportion of CoxA16 infections in the group of individuals 5–9 years of age increased by 8.15%, the proportion of severe cases increased by 2.28 times, and the proportions of infections in the rest of the target population layers also decreased. In different population groups and layers, the proportion of infections with other enteroviruses increased significantly before and after the intervention, i.e., in the 5–9-year-old group, it increased by 88.73%, in nursery children, it increased by 89.04% and it increased in the severe cases group by 1.44 times. There were statistically significant differences in the proportions of virus types before and after the intervention in different population groups and layers (*p* < 0.001), as indicated in Table 4.

### 3.3. Changes in the Incidence Rate before and after the Vaccine Intervention

During the monitoring period, from 2014 to 2021, the incidence rate of HFMD and the number of hospitalizations, severe cases and deaths decreased after the EV71 vaccine intervention, as shown in Table 5. The incidence rate of HFMD decreased from 39.02‰ in 2014 to 11.02‰ in 2021, i.e., by 71.7%, and the decrease was statistically significant (X^2^ _(trend)_ = 86,699, *p* < 0.001). The number of hospitalized cases decreased by 68.88%, the number of severe cases dropped by 95.60% and the number of deaths dropped to 0. The decreases in the number of hospitalized cases, severe cases and deaths were statistically significant (trend X^2^ _(trend)_ = 393.73, *p* < 0.001); see Table 5.

Figure 1 shows the seasonal trend and characteristics of the incidence of HFMD before and after the vaccine intervention in Fujian Province. (1) The incidence of HFMD (Plot A) showed “double peaks” in the summer and autumn, but the “double peaks” and their widths in the non-intervention group were higher than in the intervention group. That is, the incidence in the non-intervention group continued to rise at Week 10, reached a peak at Week 23 (18,231 cases) and then continued to decline, approaching a trough at Week 29 which spanned about 4 months, and then forming the second peak. In contrast, the incidence in the intervention group began to rise slowly at Week 13; after Week 19, it rose rapidly, reached a peak at Week 21 (8997 cases), slowly decreased, maintained a plateau period of about 1 month and then continued to decline after Week 26, reaching a trough. The second peak in both groups began at Week 36, and the rising rate was similar. The incidence in the intervention group reached its peak at Week 38 (8141 cases), rapidly decreased until Week 41 and then slowly trailed until about Week 50; the incidence in the non-intervention group reached its peak at Week 40 (12,436 cases) and then decreased rapidly, trailing until about Week 51. (2) To summarize the analysis of plots B, C and D, the first peak was mainly associated with the EV71 and Cox A16 viruses, and the second peak was mainly associated with the other enteroviruses in autumn. (3) The linear equation of the incidence of HFMD in the two groups (Plot A) was y_a_ = 524.69 + 0.4787X_a_. The relationship between the two variables was statistically significant (t = 14.10, *p* < 0.001); R_a_^2^ = 0.79 > 0.5. (4) The linear equation of the incidence of EV71 (Plot B) was y_b_ = 8.7929 + 0.2188X_b_. The relationship between the two variables was statistically significant (t = 4.56, *p* < 0.001); R_b_^2^ = 0.28 < 0.5. (5) The linear equation of the incidence of Cox A16 (Plot C) was y_c_ = 5.7977 + 1.3186X_c_. The relationship between the two variables was statistically significant (t = 14.43, *p* < 0.001); R_b_^2^ = 0.80 > 0.5. (6) The linear equation of the incidence of HFMD in the presence of other enteroviruses (Plot D) was y_d_ = 39.633+ 0.6909X_d_. The relationship between the two groups of variables was statistically significant (t = 12.35, *p* < 0.001); R_d_^2^ = 0.75 > 0.5.

## 4. Discussion

Based on the introduction of the EV71 vaccine into the management of non-EPI vaccines in Fujian Province since 2017, this study analyzed and compared the incidence rate and severity of HFMD (hospitalized cases, severe cases and deaths) and the changing trend of etiology in the target population over a four-year period before and after the vaccination intervention to obtain and evaluate real-world evidence of the effect of the EV71 vaccine after its initial application.

The results of the study showed that in the target population, the incidence rate and the number of hospitalized cases, severe cases and deaths in the EV71 vaccine intervention group decreased or decreased significantly (*p* < 0.001) compared with the non-intervention group. It is suggested that after the EV71 vaccine introduction, the epidemic situation of HFMD in Fujian Province has changed, and the public health emergencies caused by HFMD have been effectively controlled. The protection ratio against the disease after intervention was 47.58%, and the efficacy index was 1.91; the protection ratio against hospitalization was 41.36%, and the efficacy index was 1.71; the protection ratio against severe cases was 94.64%, and the efficacy index was 18.64; the protection ratio against death was 95.71%, and the efficacy index was 23.29. In an analysis of the comprehensive effect of HFMD caused by multiple pathogens, the results highlight the protective effect of EV71 vaccine immunization against severe disease and death caused by HFMD, which, in turn, proved that the severe disease and death caused by multi-pathogen HFMD were mainly caused by the EV71 virus. Severe and fatal cases were mostly caused by the EV71 virus. In this study, we considered that HFMD mainly harms children under 5 years of age, which is consistent with relevant reports [10,11,12,13,14]. We found that the number of HFMD cases in children aged 0–4 years accounted for 91.52% of the total number of cases, 92.85% of the hospitalized cases, 96.23% of the severe cases and 95.65% of deaths. According to the statistical results, the protection ratio against HFMD in the EV71 vaccine group aged 0–4 years was 49.58%, and the efficacy index was 1.99; the protection ratio against hospitalization was 44.20%, and the efficacy index was 1.79; the protection ratio against severe disease was 94.81%, and the efficacy index was 19.27; the protection ratio against death was 95.57%, and the efficacy index was 22.59; the proportion of cases with EV71 virus infection in HFMD-laboratory-confirmed cases decreased significantly after the intervention in the different groups, i.e., in the 0–4-year-olds, it decreased by 68.41%, in nursery children, it decreased by 74.32% and it decreased in the severe case and death groups by 86.07% and 100%, respectively. The EV71 epidemic season changed after the intervention (Figure 1A,B). The EV71 vaccine intervention changed the seasonal characteristics of EV71, while the epidemic seasons of CoxA16 and other enteroviruses did not change, with R^2^ values of 0.80 and 0.75, respectively (Figure 1C,D). These real-world data reveal that when considering the multi-pathogenic disease HFMD, the EV71 virus is the main cause of severe and fatal cases as well as of public health emergencies in kindergartens [2]. EV71 vaccine immunization focuses on protecting the population against severe and fatal HFMD and, as we demonstrated, also may play a positive role in the reduction in the overall incidence rate of HFMD and in hospitalizations. Since the EV71 vaccine is a non-EPI vaccine, the vaccination rate is not high (estimated to be 50%), and HFMD occurrence and immunization are still uneven in different regions, which may prevent the EV71 vaccine from achieving its ultimate public health effect. The EV71 vaccine is currently a non-immunization-program vaccine in China, and residents voluntarily pay for it. It is suggested that the health administration departments strengthen the diffusion of scientific information among the population so as to improve people’s willingness to vaccinate. The government should take into consideration the gradual inclusion of the EV71 vaccine into the immunization program so to improve the vaccination rate.

In this study, real-world data on the incidence level, severity, and pathogen variations in laboratory-confirmed cases of HFMD in Fujian Province were used for the first time to evaluate the changes in the epidemiological characteristics of HFMD after the vaccine application and the public health effect of EV71 vaccine immunization. Compared with the health economics evaluation report on EV71 vaccine immunization [15], it achieved relatively satisfactory results. The evaluation of a vaccine’s immune effects mainly includes the evaluation of the vaccine immune response, immune safety, immune efficacy (protection ratio based on phase III clinical trials) and immune effect. The immune effect evaluation was based on real-world data which have the highest level of medical evidence and were insufficiently studied in other vaccine applications in China in the past.

This study suggests that except for the EV71 virus, other HFMD pathogens have changed little as pathogenic hazards. The summer peak was mainly associated with the Cox A16 and EV71 viruses. After the EV71 vaccine immunization, the Cox A16 virus acquired a relatively dominant role in HFMD pathogenesis (see Plot C). The autumn peak was mainly associated with other enteroviruses, and the pathogenic advantages they acquired after the EV71 vaccine immunization are worthy of attention. The studies by Chen Wei et al. on other enteroviruses causing HFMD [16] suggest that in Fujian Province, Cox A10 is one of the main pathogens, with a high incidence rate of severe cases; CoxA10 Fujian isolates co-evolved and co-circulated with other domestic isolates, which should be taken into consideration.

HFMD is an infectious disease caused by multiple pathogens whose monitoring reflects multi-pathogen data. The EV71 vaccine is applicable to children under 5 years of age and was introduced in 2017, making it difficult to design cohort studies. This study used big data for statistical analysis and inference, which have certain limitations. Further case–control studies are necessary. A study of the spatial clustering and the evolution of HFMD in the Chinese mainland from 2008 to 2017 found that the overall spatial distribution patterns of reported incidence, severe disease rate and mortality rate were completely different. This spatial distribution suggests that similar natural environments (such as air temperature, air pressure, sunshine, rainfall, etc.) play an important role in the spread of HFMD epidemics [17]. According to previous studies [18,19], the factors related to severe cases were male children, pathogen type (mainly EV71), physical fitness, etc. Therefore, we should include more factors for consideration in our future studies.

## 5. Conclusions

HFMD is an infectious disease caused by multiple pathogens whose monitoring reflects multi-pathogen data. In this study, real-world data on the incidence level, severity, and pathogen variations in laboratory-confirmed cases of HFMD in Fujian Province were used for the first time to evaluate the changes in the epidemiological characteristics of HFMD after the vaccine application and the public health effect of EV71 vaccine immunization. However, the spatial clustering and the evolution of HFMD in the Chinese found that similar natural environments (such as air temperature, air pressure, sunshine, rainfall, etc.) play an important role in the spread of HFMD epidemics. Therefore, we should include more factors for consideration in our future studies, further case–control studies are necessary.

## Figures and Tables

**Figure 1 vaccines-11-00944-f001:**
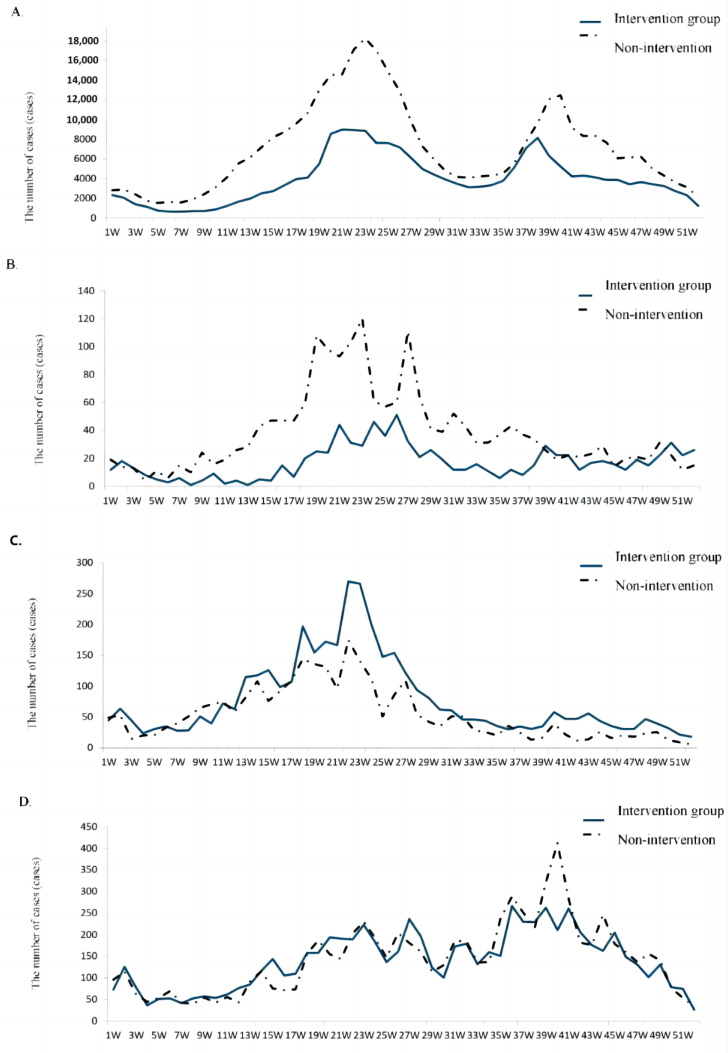
Seasonal trend of the number of HFMD cases before and after vaccination in Fujian province. (**A**) The number of HFMD cases; (**B**) comparison of the number of HFMD cases caused by EV71; (**C**) comparison of the number of HFMD cases caused by CoxA16; (**D**) comparison of the number of HFMD cases caused by other enteroviruses.

**Table 1 vaccines-11-00944-t001:** Comparison of HFMD incidence rate and number of severe cases before and after the EV71 vaccine intervention in Fujian Province.

Group	Person-Year	Case Number (N)	Incidence Rate (‰)	Hospitalized Cases (N)	Hospitalized Cases (%)	Severe Cases (N)	Severe Cases (%)	Deaths (N)
Intervention group	12,165,153	206,536	16.98	16,336	7.91	57	0.03	1
Non-intervention group	11,493,211	372,243	32.39	26,318	7.07	1004	0.27	22
Total	23,658,364	578,779	24.46	42,654	7.37	1061	0.18	23

**Table 2 vaccines-11-00944-t002:** Comparison of HFMD incidence rate and number of severe cases in the population aged 0–4 years before and after the EV71 vaccine intervention in Fujian Province.

Group	Person-Year	Case Number (N)	Incidence Rate (‰)	Hospitalized Cases (N)	Hospitalized Cases (%)	Severe Cases (N)	Severe Cases (%)	Deaths (N)
Intervention Group *	5,079,054	186,297	36.68	14,858	7.98	54	0.03	1
00-	1,067,588	24,200	22.67	1891	7.81	12	0.05	0
01-	1,047,911	66,489	63.45	5433	8.17	26	0.04	1
02-	1,068,158	40,834	38.23	3387	8.29	9	0.02	0
03-	973,339	34,193	35.13	2631	7.69	4	0.01	0
04-	922,058	20,581	22.32	1516	7.37	3	0.01	0
Non-intervention group	4,720,597	343,406	72.75	24,748	7.21	967	0.28	21
00-	982,260	52,966	53.92	4066	7.68	214	0.40	2
01-	1,004,465	127,012	126.45	10,007	7.88	397	0.31	10
02-	1,012,532	77,913	76.95	5362	6.88	184	0.24	7
03-	881,758	54,822	62.17	3529	6.44	115	0.21	1
04-	839,582	30,693	36.56	1784	5.81	57	0.19	1
Total	9,799,651	529,703	54.05	39,606	7.48	1021	0.19	22

Note: * In the intervention group and non-intervention group, 00—represents age < 1 year old; 01—represents 1 year old < age ≤ 2 years old; 02—represents 2 years old < age ≤ 3 years old; 03—represents 3 years old < age ≤ 4 years old; 04—represents 4 years old < age ≤ 5 years old.

**Table 3 vaccines-11-00944-t003:** Comparison of HFMD incidence rate and number of severe cases in the population aged 0–4 years before and after the EV71 vaccine intervention in each district and city of Fujian Province.

Region	Incidence Rate (‰)			PR (%)	Severe Cases (N)			Deaths (N)			PR * (%)
	Intervention	Non-Intervention	Total		Intervention	Non-Intervention	Total	Intervention	Non-Intervention	Total	
Fuzhou	27.42	69.30	47.86	60.43	26	513	539	0	2	2	95.19
Xiamen	46.25	61.48	53.45	24.77	3	44	47	0	2	2	94.15
Putian	19.49	41.98	30.39	53.57	3	61	64	1	4	5	94.21
Sanming	47.45	78.16	62.32	39.29	1	42	43	0	3	3	97.92
Quanzhou	30.59	61.94	44.97	50.61	0	16	16	0	1	1	100.00
Zhangzhou	44.77	83.30	63.31	46.25	8	18	26	0	2	2	62.87
Nanping	37.74	71.62	54.16	47.31	7	57	64	0	1	1	88.65
Ningde	47.14	93.98	69.87	49.84	2	44	46	0	2	2	95.90
Longyan	50.24	113.50	80.92	55.74	4	168	172	0	4	4	97.81
Pingtan	51.48	94.46	71.29	45.50	0	4	4	0	0	0	100.00
Total	36.68	72.75	54.05	49.58	54	967	1021	1	21	22	94.83

Note: * is the PR calculated from the sum of severe cases and deaths.

**Table 4 vaccines-11-00944-t004:** Changes in HFMD pathogenic dominant strains before and after the EV71 vaccine intervention in different populations in Fujian Province.

Population Group	Non-Intervention Group					Intervention Group				
	Case	Proportion	EV71	CoxA16	Others	Case	Proportion	EV71	CoxA16	Others
	(N)	(%)	(%)	(%)	(%)	(N)	(%)	(%)	(%)	(%)
Age										
0–4	13,671	94.02	26.97	19.72	53.31	9363	91.45	8.52	19.37	72.10
5–9	801	5.51	44.69	20.97	34.33	798	7.79	12.53	22.68	64.79
10–14	69	0.47	24.64	15.94	59.42	77	0.75	11.69	15.58	72.73
Gender										
Male	9255	63.65	27.75	19.79	52.46	6233	60.88	8.42	19.49	72.08
Female	5286	36.35	28.26	19.73	52.01	4005	39.12	9.54	19.78	70.69
Occupation										
Children living at home	12,336	84.84	26.10	18.40	55.50	7772	75.91	8.48	17.55	73.97
Nursery children	1985	13.65	38.09	28.61	33.30	2189	21.38	9.78	27.27	62.95
Students	220	1.51	39.09	16.82	44.09	277	2.71	12.27	16.61	71.12
Conditions										
Mild	13,805	94.94	26.08	20.51	53.41	10,196	99.59	8.89	19.60	71.52
Severe	723	4.97	62.10	5.95	31.95	41	0.40	2.44	19.51	78.05
Died	13	0.09	92.31	7.69	0.00	1	0.01	0.00	100.00	0.00
Hospitalized	14,541	100.00	27.93	19.77	52.29	10,238	100.00	8.86	19.60	71.54
Region										
Street	3690	25.38	30.73	16.69	52.57	2523	24.64	11.22	17.64	71.15
Town	9129	62.78	27.14	20.80	52.05	6154	60.11	7.78	20.62	71.60
Countryside	1588	10.92	25.57	21.54	52.90	972	9.49	2.88	21.30	75.82
Unknown	134	0.92	13.43	32.84	53.73	86.00	0.84	14.60	19.86	65.53
Total	14,541	100.00	27.93	19.77	52.29	10,238	100.00	8.86	19.60	71.54

**Table 5 vaccines-11-00944-t005:** Trend of HFMD incidence rate and number of severe cases before and after the EV71 vaccine intervention in Fujian Province.

Year	Person-Year	Case (N)	Incidence Rate (‰)	Hospitalized Cases (N)	Hospitalized Cases (%)	Severe Cases (N)	Severe Cases (%)	Deaths (N)
2014	2,846,514	111,072	39.02	8416	7.58	333	0.30	6
2015	2,909,478	85,982	29.55	6374	7.41	331	0.38	6
2016	2,859,932	87,528	30.60	5527	6.31	73	0.08	2
2017	2,877,287	87,661	30.47	6001	6.85	267	0.30	8
2018	2,958,356	94,765	32.03	7419	7.83	28	0.03	0
2019	3,036,193	51,013	16.80	4113	8.06	10	0.02	1
2020	3,085,302	26,682	8.65	2185	8.19	4	0.01	0
2021	3,091,611	34,076	11.02	2619	7.69	15	0.04	0

## Data Availability

The datasets used in the present study are available from the corresponding author (dongj8888@163.com) on reasonable request.

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
