# Peer review of "Immune Efficacy of the EV71 Vaccine in Fujian Province, China: A Real-World Analysis of HFMD"

_vaccines, 2023, doi:10.3390/vaccines11050944_

Round 1
Reviewer 1 Report (Previous Reviewer 3)
I was invited to revieww the resubmitted version of the paper entitled "Immune Efficacy of the EV71 Vaccine in Fujian Province, China: A Real-World Analysis on HFMD". The aim of thi study was to evaluate the impact of the EV71 vaccination against hand-mouth-foot disease in a province of China.
Despite several revisions, many criticism are still present.
- Introduction section was too poor. In addition some statement reported poor references. In particular, Authors stated "a small number of children may suffer from complications such as myocarditis, pulmonary edema, and aseptic meningoencephalitis. If the disease develops rapidly, as in severe cases, HFMD may lead to death", but the references referred to this sentece was old (2007) and describe several type of enteroviruses and was not focused on the main objective of this paper;
- Lines 33-34 was not supported by references;
- About methods, the sources of data has to be deeply described, reporting also which variables were included;
- About enrolled patients, it is unclear if only full vaccinated patients were included;
- Incidence rate shoulb be standardized for age and gender according to Fujian population;
- It is totally unclear how Authors evaluate the follow-up period (person-year). This point was already raised in prior revision and it was not addressed. In particular, no certain data about the followup of patients were reported. It is unclear if Authors have single patients data or aggregate data;
- If Authors decided to perform a cohort study, a survival model should be performed. Linear regression was not appropriate to evaluate this kind of trend. If Authors did not have single patients data, they should perform a more appropriate statistical methods (Jointpoint regression?);
- Discussions were poor, poor references were reported;
- Strenght and limitation section should be improved.
Author Response
Thank you for your valuable review comments. We provide a point-by-point response as follow:
1- Introduction section was too poor. In addition some statement reported poor references. In particular, Authors stated "a small number of children may suffer from complications such as myocarditis, pulmonary edema, and aseptic meningoencephalitis. If the disease develops rapidly, as in severe cases, HFMD may lead to death", but the references referred to this sentece was old (2007) and describe several type of enteroviruses and was not focused on the main objective of this paper;
Reply: Based on newer studies, we changed this expression that "a small number of children may suffer from complications such as myocarditis, pulmonary edema, and aseptic meningoencephalitis. If the disease develops rapidly, as in severe cases, HFMD may lead to death ", replaced with "Human enterovirus barely causes gastrointestinal symptoms such as gastroenteritis, but does cause diseases with a variety of clinical symptoms, children infected with human enterovirus are susceptible to hand, foot and mouth disease. There is a certain probability of aseptic meningitis, meningoencephalitis, acute respiratory diseases, acute myocarditis and acute delayed paralysis. The reference is: Muir P. Enteroviruses. Medicine, 2014, 42(1):57-59.
2- Lines 33-34 was not supported by references;
Reply: We added a relevant reference. Weiyi Pan, Zhiyuan Zhao and Junlei Chen. Birth cohort analysis on inoculation rate of enterovirus 71 vaccine among children at ages of 0 ~ 5 years in Fujian Province,China from 2017 to 2018[J]. Chin J Biologicals, 2020, 33(3): 293-6.
As mentioned in this reference, in 2016, the EV71 vaccine was officially launched, which is independently developed by China. In 2017, EV71 vaccine was included in the "Fujian Province Second Class Vaccine centralized Procurement Catalog", providing a new method for preventing HFMD and reducing the occurrence of severe illness and death in Fujian Province.
3 - About methods, the sources of data has to be deeply described, reporting also which variables were included
Reply: The case information of HFMD was obtained from the infectious disease reporting information management system of Chinese Disease Control and Prevention Information System (Network Direct reporting System), and the study target population was obtained from the network direct reporting system "Basic Information System". The network direct reporting system shall be carried out by the doctors of township hospitals, maternal and child health care hospitals and hospitals within 24 hours after the disease is diagnosed as HFMD, and the network direct reporting and case report card shall be filled in. If there is any modification or deletion, the reasons shall be stated. The reported information needs to be reviewed at the county, city, and provincial CDC levels. In addition, the disease control and prevention department will check the inpatient records, outpatient cases and case report cards for many times in each health center and hospital within a year to see whether there is missing or wrong report, and issue the corresponding report. Report variables included name, gender, date of birth, contact telephone number, address of current residence, population classification, case classification, date of onset, diagnosis time, date of death, disease name, disease type before revision, diagnosis time before revision, final review time before revision, doctor filling card, doctor filling card date, reporting unit area code, reporting unit, unit type, report card entry time and record card User, unit to which the card recording user belongs, county and district audit time, city and prefecture audit time, province and city audit time, audit status, revised report time, revised final audit time, final audit death time, revised user, unit to which the revised user belongs, etc. It can also be seen from the reported variables that the network direct reporting system is relatively complete and can guarantee the authenticity of data.
4- About enrolled patients, it is unclear if only full vaccinated patients were included;
Reply: Not only those who had been fully vaccinated, but also those who had not been vaccinated or had not been fully vaccinated. The target population of this study includes people under 15 years of age (0-14 years of age) observed annually during the monitoring period from 2014 to 2021. Since EV71 vaccine was included in the management of non-immunization program vaccine in 2017 in Fujian Province, the design idea of group was adopted in this study, so that the target population in the four years after EV71 vaccine intervention (2018-2021) was the intervention group, and the number of included cases was 206536. During the first four years of intervention (2014-2017), the target population was the non-intervention group, with 372,243 cases included. A total of 578,779 cases were included. That is, this study was conducted to study the immunizations of people in the real world before and after the change of immunization policy.
5 - Incidence rate should be standardized for age and gender according to Fujian population;
Reply: As this study is a study on the real-world immune effect of EV71 vaccine application population in Fujian Province, the intervention group and non-intervention group in this study belong to children aged 0-14 years in Fujian Province during 2018-2021 and 2014-2017 respectively, that is, all belong to the whole population in the real world. In addition, this study is also committed to the study of the effect of EV71 vaccine applied in the real world, so it should not be necessary to standardize the incidence of age and sex according to the Fujian population.
6- It is totally unclear how Authors evaluate the follow-up period (person-year). This point was already raised in prior revision and it was not addressed. In particular, no certain data about the followup of patients were reported. It is unclear if Authors have single patients data or aggregate data;
Reply: Due to the large sample size of this study, we used the approximate method to calculate the exposed person years, that is, the average number of people multiplied by the number of observation years to get the total number of people years, and the average number was the number of people in the middle of the year. We have case-by-case data.
7-If Authors decided to perform a cohort study, a survival model should be performed. Linear regression was not appropriate to evaluate this kind of trend. If Authors did not have single patients data, they should perform a more appropriate statistical methods (Jointpoint regression?);
Reply: The purpose of using linear regression is not to assess the trend, so there should be no need to establish a survival model in this study. Pearson correlation analysis was used to obtain R2 value (R2≥0.5 indicates that the linear relationship between the two quantitative variables is very close and has certain use value), so as to understand whether the dependence relationship between variables is statistically significant.
8 - Discussions were poor, poor references were reported;
Reply: We revised the article more strictly.
9 - Strength and limitation section should be improved.
Reply: We revised the article more strictly.
Reviewer 2 Report (Previous Reviewer 2)
In the paper, the authors have compared the incidence rate, severity and etiological changes of HFMD in a target population before and after the vaccine intervention. The incidence rate of HFMD decreased from 39.02‰ in 2014 to 11.02‰ in 2021, with a decrease rate of 71.7%, and the decrease was statistically significant (p < 0.001). The number of hospitalized cases decreased by 68.88%, that of severe cases dropped by 95.60%, and that of deathsdropped to 0. The proportion of cases caused by the EV71 virus in different populations decreased significantly after the intervention, specifically, by 68.41% among the 0–4-year-olds, by 74.32% among kindergarten children, by 86.07% as regards severe cases and by 100% as regards the numberof deaths.
The study is well-designed and the manuscript well written, with minor stylistic flows.
Minor point:
Please explain better Table 2 subgroupings
Author Response
Thank you for your valuable review comments. Our reply as follows:
Please explain better Table 2 subgroupings.
Reply: In the intervention group and non-intervention group, 00- represents age<1 year old, 01- for 1 year old <Age ≤ 2 year old, 02 - represents 2 year old <Age ≤ 3 year old, 03 - represents 3 year old < Age ≤ 4 year old, 04 - represents 4 year old <Age≤5 year old.
Reviewer 3 Report (Previous Reviewer 1)
This is an important, well analysed and well presented study.
Analysis
You described your statistical approach thus:
“The frequency data were inferred by X2 or trend X2 , and p < 0.05 was considered sta- 83 tistically significant. A protection ratio (PR) ≥ 50% indicated that the vaccine was effective. 84 Analysis of the correlation or dependence between two quantitative variables[5]. (1) Use 85 simple linear regression analysis to obtain a straight line that passes through the origin of 86 the rectangular coordinate system and use the t test or F test to compare the difference 87 between the slope and 0; if p < 0.05, the correlation between the two variables is statisti- 88 cally significant; (2) Pearson correlation analysis was used to obtain R2 ≥ 0.5, indicating 89 that the correlation between the two quantitative variables was highly linear, and the 90 Vaccines 2022, 10, x FOR PEER REVIEW 3 of 9 result was meaningful. The conversion formula of Box-Cox skewness data was used for 91 the incidence[6], (Z= ï¼¹ λ λ , λ≠0) gradually adjusted and determined by the Kolmogorov– 92 Smirnov normality test.”
[Please state more clearly why you chose these statistical techniques as suitable for your data and how they “were gradually adjusted and determined by the Kolmogorov– 92 Smirnov normality test”]
Results.
It is not surprising that the vaccine exerted selective pressure on vaccine virus types
You wrote: “The differences in the proportions of virus types 166 before and after the intervention were statistically significant (X2 = 1459.52, p < 0.001). In 167 different population groups and layers, the proportions of EV71 virus infections de- 168 creased significantly before and after the intervention, specifically, by 68.41% in 0–4-year- 169 olds, by 74.32% in nursery children, by 86.07% as regards severe cases and by 100% as 170 regards deaths. The proportion of CoxA16 infections in the 5–9-year-old group increased 171 by 8.15%, the proportion of severe cases increased by 2.28 times, and the proportions of 172 infections in the rest of the target population layers also decreased. In different population 173 groups and layers, the proportion of infections with other enteroviruses increased signif- 174 icantly before and after the intervention, i.e., in the 5–9-year-old group, it increased by 175 88.73%, in nursery children by 89.04%, and in the severe cases group by 1.44 times. There 176 were statistically significant differences in the proportions of virus types before and after 177 the intervention in different population groups and layers (p < 0.001), as indicated in Table 178 4.”
[Do you have data why some children had more severe cases or died? Immunodeficiency, malignancies, living conditions, sibs with HFMD or other illness…?]
Do you have ideas why rates differed between residence areas?]
Author Response
Thank you for your valuable review comments. Our responses as follows:
1 Please state more clearly why you chose these statistical techniques as suitable for your data and how they “were gradually adjusted and determined by the Kolmogorov– 92 Smirnov normality test”
Reply: By comparing the maximum absolute difference between the cumulative distribution function of sample data and the theoretically assumed cumulative distribution function of normal distribution, we use K-S normal test to test whether the samples in this study conform to the normal distribution, so as to determine the validity of sample sampling information.
2 Do you have data why some children had more severe cases or died? Immunodeficiency, malignancies, living conditions, sibs with HFMD or other illness…? Do you have ideas why rates differed between residence areas?
Reply: The main objective of this study was to understand the real-world immune effects of EV71 vaccine in Fujian Province. No research has been carried out on the problems raised by the review opinions. Only a few other findings can be shared here. The main pathogenic enterviruses of HFMD were Enterovirus 71 (EV71) and Coxsackievirus A-16 (CVA16). Other human enterviruses (non-EV71 and non-CVA16 human enterovirus) accounted for A certain proportion. The pathogenicity of different types of enterovirus is different, so it may be partly responsible for the difference in the illness of infected people. According to previous studies, the factors related to severe cases are male children, pathogen type (mainly EV71), physical fitness, etc.
A study on the spatial aggregation and evolution of hand-foot-mouth disease in mainland China from 2008 to 2017 found that in the Moran scatter plot, the incidence of hand-foot-mouth disease in Hainan, Guangdong, Guangxi, Hunan, Zhejiang, Fujian and Shanghai was in the first quadrant (high) for most of the time. The rate of severe diseases in Hainan, Guangdong, Yunnan, Guizhou, Hunan and Henan was in the first quadrant (high) or the fourth quadrant (high). The mortality rate of HFMD in Hainan, Guangdong, Chongqing, Guizhou and Hunan was in the first quadrant (high). The overall spatial distribution patterns of reported morbidity, severe morbidity and mortality of hand, foot and mouth disease from 2008 to 2017 were not identical. This spatial distribution suggests that similar natural environments (such as temperature, air pressure, sunshine, rainfall, etc.) play an important role in the spread and prevalence of HFMD. Hand-foot-mouth disease (HFMD) has various transmission routes, most of which are recessive infections [9], with more than 20 pathogens. The change of dominant pathogens is easy to cause the spread of the disease, which is an important reason for the spatial clustering of the incidence.
Round 2
Reviewer 1 Report (Previous Reviewer 3)
- Lines 106-114 are totally unreadable. The description of Pearson correlation meaning is totally useless for the reader;
- Authors should report an impact measure to show the effect of vaccination. For example attributable risk;
Author Response
Response to Reviewer 1 Comments
Point 1: Lines 106-114 are totally unreadable. The description of Pearson correlation meaning is totally useless for the reader
Response 1: 1.Our use of Pearson's correlation analysis was to understand whether there was an association between the intervention group and the non-intervention group and its strength of association. The description of this paragraph is a valuable opinion from another reviewer expert, so we made the following description of the research method.
Point 2: Authors should report an impact measure to show the effect of vaccination. For example attributable risk
Response 2: Thank you for your valuable advice, but we have done the protection rate (PR) and effect index (IE), where the algorithm and results are the same as attributable risk (AR) and relative risk (RR). So we don't necessarily need to add AR and RR. Calculated based on the values presented in Table 1
Table 1. Comparison of HFMD incidence rate and number of severe cases before and after the EV71 vaccine intervention in Fujian Province.
|
Group |
Person-Year |
Case Number (N) |
Incidence Rate (‰) |
Hospitalized Cases (N) |
Hospitalized Cases (%) |
Severe Cases (N) |
Severe Cases (%) |
Deaths (N) |
|
Intervention group |
12,165,153 |
206,536 |
16.98 |
16,336 |
7.91 |
57 |
0.03 |
1 |
|
Non-intervention group |
11,493,211 |
372,243 |
32.39 |
26,318 |
7.07 |
1004 |
0.27 |
22 |
|
Total |
23,658,364 |
578,779 |
24.46 |
42,654 |
7.37 |
1061 |
0.18 |
23 |
1.The attributable risk (AR) is the absolute value of the difference between the incidence between the exposed group and the non-exposed group, indicating that the extent of risk is specifically attributable to the cause of exposure, that is, the increase or decrease of the incidence of the exposure group.
AR%= (cumulative incidence (or mortality) in the exposure group-cumulative incidence (or mortality) in the control group) / cumulative incidence (or mortality) in the exposure group * 100%
AR% (incidence rate) = (32.39 ‰ -16.98 ‰) / 32.39 ‰ *100%=47.58%
This indicates that the proportion of the EV71 vaccine risk in the nonintervention group was 47.58%.
2.Relative risk (RR) indicates how many times the morbidity or mortality in the exposure group is the control group. It indicates that the risk of morbidity or death in the exposed group is a multiple of the non-exposed group. A larger RR value would indicate a greater effect of exposure and the strength of the association of exposure with outcome. Its numerical significance[1]
1.RR was 0.9~1 or 1.0~1.1, indicating that exposure factors were not associated with disease;
- The RR was 0.7~0.8 or 1.2~1.4, indicating a weak association between exposure factors and disease;
- The RR was 0.4~0.6 or 1.5~2.9, indicating the association between exposure factors and disease;
- The RR was 0.1-0.3 or 3.0-9.9, indicating a strong association between exposure factors and disease;
- The RR is less than 0.1 or greater than 10, indicating a strong association of the exposure factors with the disease.
RR= cumulative incidence (or mortality) in exposure group / cumulative incidence (or mortality) in control group
RR (incidence) =32.39 ‰ / 16.98 ‰ =1.91, indicating that the incidence of the non-intervention group was 1.91 times that of the intervention group, and EV71 vaccination was moderately associated with HFMD.
[1] Li Liming. epidemiology. Beijing. The People's Health Publishing House. 2008.11.
A description of the protection ratio (PR) and the effect index is given in lines 99-106.
“(2) Protection ratio (PR):represent the annual incidence rate of HFMD in the intervention group and the non-intervention group, respectively. Combined with the EV71 pathogen proportion, it reflects the ratio of protection against HFMD or death in the EV71 vaccine immune intervention population relative to the non-immunized population.
(3) Efficacy Index(IE).It indicates the proportion of incidence or mortality rate in the non-intervention group and the intervention group and indirectly reflects the effect of the EV71 vaccine in the immunized population.”
A description of the effect index and the protection rate is available in line 164 of the article.
“The statistical analysis showed that the protection ratio of the two groups before and after the EV71 vaccine intervention was 47.58%, and the efficacy index was 1.91; “
This manuscript is a resubmission of an earlier submission. The following is a list of the peer review reports and author responses from that submission.
Round 1
Reviewer 1 Report
Thank you for this important, excellently researched and presented study.
Methods
“Fujian Province has long implemented an infectious disease underreporting and reporting quality inspection system, as well as a population and system code maintenance review meeting system. In addition to the duplicate inspection of case records within a natural year in the online direct reporting system, the duplicate inspection and data cleaning of aggregated database were carried out again according to "Current address GB", "Name", "Gender", "Date of birth" four indicators across annual case records to ensure accuracy of monitoring and observation records.”
{please describe the underreporting methods. This is crucial to your results and will be of great interest to other researchers]
Results
“From 2014 to 2021, the number of HFMD cases aged 0-4 years in Fujian Province was 116 529,703, accounting for 91.52% of the total cases; the incidence rate was 54.05‰. The number of hospitalized cases was 39,606, accounting for 92.85% of the total number of hospitalizations, the number of severe cases was 1,021, accounting for 96.23% of the total number of severe cases, and 22 people died, accounting for 95.65% of the total number of 120 deaths.”
“During the surveillance period from 2014 to 2021, there were 24,779 laboratory-con- 143 firmed cases of HFMD, all of which were hospitalized.”
[It appears that none of the community cases that were not hospitalised had laboratory proven EV71. Is this policy?]
[Table 3 Can you please comment on differences between regions in incidence rates?]
Conclusions
“This study suggests that, except for EV71 virus, other HFMD pathogens have little change in the pathogenic hazards. The summer peak is mainly composed of Cox A16 and 245 EV71 viruses. After EV71 vaccine immunization, Cox A16 virus has a relatively dominant role in pathogenesis (see Plot C). The autumn peak is mainly caused by other enteroviruses, and its pathogenic advantages are more worthy of attention after EV71 vaccine im- 248 munization. Chen Wei et al[7] studies on other enteroviruses in HFMD suggest that other enteroviruses in HFMD in Fujian Province have Cox Al0 as one of the main pathogens, with a high incidence rate of severe cases; CoxA10 Fujian isolates co-evolved and co-cir-culated with other domestic isolates.”
You do not make any suggestions how to increase vaccination rates to decrease the rate incidence further. What do you recommend to the Ministry of Health please?]
Author Response
1. please describe the underreporting methods. This is crucial to your results and will be of great interest to other researchers
Every year, the health administration department of Fujian Province shall organize an investigation on the underreporting and reports quality of infectious diseases, proactively review the registration forms of all medical institutions' visits and inpatient cases, and timely make a supplementary report if any underreporting cases are found.
2. It appears that none of the community cases that were not hospitalised had laboratory proven EV71. Is this policy?
This is not a policy. As for now, laboratory proven EV71 can only be confirmed in general hospitals and provincial CDC laboratories, the accessibility of community laboratories is not enough.
3. You do not make any suggestions how to increase vaccination rates to decrease the rate incidence further. What do you recommend to the Ministry of Health please?
EV71 vaccine is currently a non-immunization program vaccine in China, and residents voluntarily pay for it. It is suggested that the health administration departments strengthen the publicity of science popularization and improve the people's willingness to vaccinate. And the government should gradually include EV71 vaccine into the immunization program in areas to improve the vaccination rate.
Reviewer 2 Report
In this paper the authors have collected data from 4 years, in order to compare the incidence rate, severity and etiological changes of HFMD between the target population before and after the EV71 vaccine intervention. The incidence rate of HFMD decreased from 39.02‰ in 2014 to 11.02‰ in 2021 (71.7% decrease); The number of hospitalized cases decreased by 68.88%, the severe cases dropped by 95.60%, and deaths dropped to 0. The proportion of cases caused by EV71 virus in different populations decreased significantly after intervention, which decreased by 68.41% in 0-4 years old, 74.32% in kindergarten children, 86.07% in severe cases and 100% in death.
The study is well-designed, involved a significant number of subjects and the description of the results is clear. The graphs are properly presented and the statistical analysis is correct.
Author Response
Thank you for your comments and suggestions to us.
Reviewer 3 Report
I was invited to revise the paper entitled "Immune Efficacy of EV71 Vaccine in Fujian Province of China: A Real-world Analysis on HFMD". It aimed to evaluate the real life efficacy of the EV71 vaccine against hand, foot and mouth disease. The topic is interesting but I have some concern about methodology.
Authors decide to compare the incidence rate of HFMD before and after the introduction of the vaccination.
- Authors did not reported the exact number of patients included during the two study period;
- Methodology is not appropriate. The incidence cannot be analyzed with a linear regression analysis. Authors should perform an appropriate analysis with the suggestion of a expert statistician. In addition, the statistical analysis section was also poorly described;
- It is unknown how Authors estimates the person-year of the two cohorts. How did they evaluated the enrollment of each patients;
- References are poor;
- Discussion section should deeply describe study results. No limitations were reported. No comparison with the literature was performed;
- English language needs to be improved.
- It is unkwon the referred year of figure 1;
Author Response
1. Authors did not reported the exact number of patients included during the two study period.
Intervention group: 12165153 person-year
Non-intervention group: 11493211 person-year
2. Methodology is not appropriate. The incidence cannot be analyzed with a linear regression analysis. Authors should perform an appropriate analysis with the suggestion of a expert statistician. In addition, the statistical analysis section was also poorly described
Box-Cox transformation (a transformation family formula) was used for incidence skewness.
Please see the attachment.
3. It is unknown how Authors estimates the person-year of the two cohorts. How did they evaluated the enrollment of each patients
Based on the official annual population, and the composition of the age groups, a programmatic calculation produces estimates of the annual population of these two groups.
Please see the attachment.
4. References are poor;
There aren't many studies of this kind.
5. Discussion section should deeply describe study results. No limitations were reported. No comparison with the literature was performed
This study used big data for statistical analysis and inference, which has certain limitations. We hope to conduct further case-control studies in the future.
6. It is unkwon the referred year of figure 1
Figure 1 shows the distribution of the total number of weekly cases during 2014-2021.

Round 2
Reviewer 3 Report
Authors did not addressed all comments. The response is not satisfactory.
Author Response
- Authors did not reported the exact number of patients included during the two study period;
Study target population: A total of 23,658,364 people under the age of 15 (0-14 years old) with the current address in Fujian Province will be observed every year during the monitoring period from 2014 to 2021. The EV71 vaccine included in the management of non-immunization program vaccine from 2017 in Fujian, a total of 2,177,931 doses of EV71 vaccine were given to children under 5 from 2018 to 2021, with a full vaccination rate of 54.82%. The group design was adopted, that the target population in the four years after EV71 vaccine intervention (2018-2021) was the intervention group, and the number of included cases was 206,536. During the first four years of vaccine intervention (2014-2017), the target population was the non-intervention group, with 372243 cases included. A total of 578,779 cases were included.
Intervention group: 12165153 person-year
Non-intervention group: 11493211 person-year
- Methodology is not appropriate. The incidence cannot be analyzed with a linear regression analysis. Authors should perform an appropriate analysis with the suggestion of a expert statistician. In addition, the statistical analysis section was also poorly described;
Box-Cox transformation (a transformation family formula) was used for incidence skewness.
Please see the attachment.
- It is unknown how Authors estimates the person-year of the two cohorts. How did they evaluated the enrollment of each patients;
Based on the official annual population, and the composition of the age groups, a programmatic calculation produces estimates of the annual population of these two groups.
Please see the attachment.
- References are poor;
There aren't many studies of this kind.
- Discussion section should deeply describe study results. No limitations were reported. No comparison with the literature was performed;
This study used big data for statistical analysis and inference, which has certain limitations. We hope to conduct further case-control studies in the future. Changes have been made in the article.
- English language needs to be improved.
Changes have been made in the article.
- It is unkwon the referred year of figure 1;
Figure 1 shows the distribution of the total number of weekly cases during 2014-2021.
